# Thoracic Aorta: Anatomy and Pathology

**DOI:** 10.3390/diagnostics13132166

**Published:** 2023-06-25

**Authors:** Cira Rosaria Tiziana di Gioia, Andrea Ascione, Raffaella Carletti, Carla Giordano

**Affiliations:** Department of Radiology, Oncology and Pathology, Sapienza, University of Rome, Viale Regina Elena 324, 00161 Rome, Italy; cira.digioia@uniroma1.it (C.R.T.d.G.); andrea.ascione@uniroma1.it (A.A.); carlettiraffaella@gmail.com (R.C.)

**Keywords:** thoracic aorta, embryology, aortic aneurysm, aortic dissection

## Abstract

The aorta is the largest elastic artery in the human body and is classically divided into two anatomical segments, the thoracic and the abdominal aorta, separated by the diaphragm. The thoracic aorta includes the aortic root, the ascending aorta, the arch, and the descending aorta. The aorta’s elastic properties depend on its wall structure, composed of three distinct histologic layers: intima, media, and adventitia. The different aortic segments show different embryological and anatomical features, which account for their different physiological properties and impact the occurrence and natural history of congenital and acquired diseases that develop herein. Diseases of the thoracic aorta may present either as a chronic, often asymptomatic disorder or as acute life-threatening conditions, i.e., acute aortic syndromes, and are usually associated with states that increase wall stress and alter the structure of the aortic wall. This review aims to provide an update on the disease of the thoracic aorta, focusing on the morphological substrates and clinicopathological correlations. Information on anatomy and embryology will also be provided.

## 1. Introduction

The aorta is the largest elastic artery in the human body and is classically divided into two anatomical segments, the thoracic aorta (TA) and the abdominal aorta (AA), separated by the diaphragm. The different segments show distinctive physiological properties reflecting their different anatomical structure and embryology [1]. The TA presents higher compliance as compared to the AA and its elastic capacity, especially in the proximal segments, actively contributes to maintaining diastolic pressure and blood flow at the level of peripheral circulation.

The heterogeneous histogenesis of the different segments of the aorta impacts the onset and natural history of congenital and acquired diseases that develop herein. The latter may present as a chronic, often asymptomatic disorder (i.e., thoracic and abdominal aortic aneurisms, TAA and AAA) or as an acute life-threatening condition i.e., acute aortic syndromes (AAS). The present review aims to provide an update on the disease of the thoracic aorta, focusing on the morphological substrates and clinicopathological correlations. Information on anatomy and embryology will also be provided.

## 2. Topographic Anatomy of the Thoracic Aorta

The thoracic aorta includes the aortic root, the ascending tract, the arch, and the descending aorta (Figure 1A–C) [2].

The aortic root extends from the annulus to the sinotubular junction (STJ), and comprises the sinuses of Valsalva, with the aortic cusps, the coronary origins, and the intercuspal triangles (Figure 1C) [3,4]. The annulus is a virtual circular line running through the base of the aortic cusps. It lies distal both to the anatomic ventricular-aortic junction (where cardiac myocytes leave the place to smooth muscle cells of the aortic wall) and the hemodynamic junction, identified by the crown-like attachment of the aortic cusps to the sinuses wall. The STJ is a virtual circular line running through the tip of the aortic valve commissures. The wall of the sinuses of Valsalva is thinner as compared to the wall of the aorta (about 2 mm versus 4 mm).

The ascending aorta comprises the tract from the STJ to approximately the level of the fourth thoracic vertebra, where the brachiocephalic artery takes off. The aortic arch lies between the brachiocephalic artery and the isthmus, distal to the left subclavian artery origin, and gives rise to the brachiocephalic, the left common carotid, and the left subclavian arteries. The descending aorta starts after the take-off of the left subclavian artery and has a thoracic and an abdominal segment.

## 3. Histology and Histogenesis of the Aorta

The aorta’s elastic properties depend on its wall structure, composed of three distinct histologic layers: intima, media, and adventitia (Figure 1B). The media is the largest component and is formed by concentrically organized lamellar units. Each lamellar unit is composed of two layers of elastic laminae and smooth muscle cells (SMCs), collagen fibers, and proteoglycans lining in between (Figure 1B, insert). The number and thickness of the lamellar units of the media increase with age and, in adults, vary according to topographic location [5]. During postnatal growth, the increase in medial thickness in the ascending aorta and arch is achieved through an increase in the number of lamellae (up to 60 in adulthood), whereas in the descending aorta through the thickening of the existing lamellar units (up to 28–30). This different behavior reflects the different embryological origins of SMCs at different topographic sites (Figure 1A), which in turn condition the response of SMCs to growth factor stimuli (e.g., TGF-β). The SMCs of the ascending aorta and the arch derive from the neural crests (i.e., from the ectoderm), while those of the descending aorta derive from the somites of the paraxial mesoderm [6,7,8]. Of note, the ratio between the aortic diameter and the medial thickness remains constant in the different segments of the aorta [9]. Thanks to the higher number of lamellar units, the TA is more compliant as compared to the AA.

The aorta does not contain a distinct inner or outer elastic lamina. At birth, the intima is thin and consists of the endothelium alone, in close contact with the first elastic lamella. In the adult, the accumulation of extracellular matrix proteins and mesenchymal cells induces an increase in intimal thickness (Figure 1B). The adventitia comprises loose connective tissue, vasa vasorum, and lymphatic vessels. Vasa vasorum normally extends into the outer third of the media.

## 4. Embryology of the Aorta

### 4.1. Embryology of the Aortic Root

The aortic root development is strictly connected with the embryology of the cardiac outflow tract (Figure 2). In midweek 5 of embryogenesis, cells from the neural crest migrate in the truncus arteriosus (the cranial portion of the primitive heart tube) and contribute to the formation of a spiral conotruncal septum, which separates the aortic and pulmonary outflow tracts [8]. From the wall of each tract, three endocardial swellings give rise to the cusps of the semilunar valves of the aorta and pulmonary artery. Excavation of truncal tissue inferior to the aortic swellings leads to the formation of the sinus of Valsalva.

### 4.2. Embryology of the Ascending Aorta, Aortic Arch, and Descending Aorta

During the third week of gestation, isolated vascular islands merge forming the paired primitive aortae. Each primitive aorta consists of a ventral and a dorsal segment in continuity through the first aortic arch. The two ventral aortae are connected to the aortic sac, an expansion at the cranial end of the truncus arteriosus [10,11,12,13,14].

The dorsal aortae extend for the entire length of the embryo and, during the 4th week, fuse from the fourth thoracic segment to the fourth lumbar segment forming the midline dorsal aorta. The latter will form the descending aorta.

By the 5th week, aortic arches 2, 3, 4, and 6 develop from the aortic sac connecting to the dorsal aortae. The first two arches regress as the later arches form (Figure 2A).

The common carotid and proximal portions of the internal carotid arteries originate from the third pair of aortic arches. The distal portion of the internal carotid arteries arises from the cranial end of the dorsal aortae (Figure 2B). The development of the fourth pair of arches is asymmetric. On the right side, the fourth arch forms the proximal portion of the right subclavian artery. The distal portion of the right subclavian artery derives from the right dorsal aorta. The aortic sac forms the brachiocephalic trunk and the first portion of the aortic arch. On the left side, the fourth arch becomes the arch of the aorta. The sixth arch contributes to the formation of the main pulmonary artery, its branches, and ductus arteriosus. Here, neural crest cells play a pivotal role in the formation of the aorticopulmonary septum, which fuses with the conotruncal septum separating the aortic and pulmonary channels [8]. On the left side, the distal portion of the left arch remains in communication with the dorsal aorta, forming the ductus arteriosus. The ductus arteriosus closes at birth and is later transformed into the ligamentum arteriosum, which attaches the pulmonary trunk to the aorta (Figure 2C).

## 5. Congenital Disease of the Aorta

### 5.1. Aortic Root Malformations

Abnormal septation of the truncus arteriosus leads to complex outflow cardiac malformation (including truncus arteriosus, tetralogy of Fallot, transposition of the great arteries, etc.) that will not be discussed in the present review.

A special mention is deserved of the bicuspid aortic valve (BAV), a congenital valvulo-aortopathy inherited in an autosomal dominant pattern with incomplete penetrance [15]. BAV is observed in 1–2% of the population, more commonly in males [16,17]. Recently, the International Consensus Classification and Nomenclature for the congenital BAV recognized three main morphological variants based on cusp size and the number of sinuses, implying different embryological origins [18]. The most common is the fused BAV type, characterized by three sinuses and two cusps, one larger than the other, which may derive either from the failure of one commissure to develop or from the fusion of two cusps during fetal life (most often right-left cusp fusion). A fibrous ridge (i.e., a raphe) is often present between the fused cusps. A milder embryological defect is the partial fusion BAV, which represents the second morphologic variant, characterized by three cusps, two of which are partially fused at the base of a commissure, with a small raphe. The third form is the 2-sinus BAV type, with two symmetric cusps without raphe, and two sinuses. It appears as the most severe embryological defect, derived by abnormal endocardial cushion formation or positioning [18]. Though usually asymptomatic in childhood, with aging, BAV undergoes fibrosis and calcification, with significant aortic stenosis and/or regurgitation and associated aortic dilation [17]. Both valvulopathy and aortopathy show heterogeneous clinical phenotypes, recently classified into three clinical-prognostic groups: (i) complex valvulo-aortopathy, where BAV is associated with other cardiovascular malformations or genetic disorders (i.e., coarctation of the aorta, Turner syndrome, Loeys–Dietz syndrome, etc.) and/or there is early valve dysfunction and/or aortopathy, diagnosed in pediatric and young adulthood; (ii) typical valvulo-aortopathy, with progressive BAV dysfunction and/or aorta dilation without concomitant disorders. This is the most common group, usually diagnosed in adulthood; (iii) undiagnosed or uncomplicated BAV, a lifelong silent condition with mild or non-progressing valvulo-aortopathy that does not manifest clinically. Both patients with complex and typical presentations are at risk of developing infective endocarditis and acute aortic dissection (AAD), although the latter is rare without aortic dilation [18]. See the paragraph on TAA etiology for details on the BAV-related aortopathy phenotype.

### 5.2. Aortic Arch Malformations and Coarctation of the Aorta

Aortic arch malformations result from the persistence or inappropriate regression of primitive aortic arches. They may be isolated or associated with complex congenital heart disease. When isolated, they may be asymptomatic and only incidentally noticed.

Among aortic arch malformations, vascular rings are rare congenital anomalies (about 1% of all cardiovascular anomalies) in which the aortic arch and its branches encircle and compress the trachea, the esophagus, or both [19,20]. According to the degree of tracheoesophageal compression, they may cause respiratory or gastroesophageal symptoms. Backer and Mavroudis proposed a classification of vascular rings into four main categories [21], although many rare variants can be observed [22]. The most frequent vascular ring (30–50% of cases) is the double aortic arch, due to the persistence of the right and left fourth arches. The arches completely encase the trachea and esophagus and join posteriorly to form the descending thoracic aorta. Respiratory distress and feeding problems usually develop during the first month of life. Associated cardiovascular anomalies are uncommon. The second most frequent type of vascular ring (12–25%) is the right aortic arch with the left ductus arteriosus and retro-esophageal aberrant left subclavian artery. The ductus arteriosus arises from the left subclavian artery. The latter may present an aneurysmal dilation at the base, called a Kommerell diverticulum. Most patients are asymptomatic, but feeding difficulties may arise when solid foods are introduced. Fallot’s tetralogy or other conotruncal anomalies may coexist [19,20]. Finally, the coarctation of the aorta deserves a rapid mention. It consists of a severe constriction of the aortic arch due to abnormal thickening of the wall. It occurs in approximately 0.3% of all live-born children, is more common in males, and is the most common cardiovascular defect in Turner syndrome. It can be associated with BAV. Clinical effects depend on the degree and site of narrowing. The most common form is the juxta-ductal (located posterior and adjacent to the insertion of the ductus arteriosus), less frequent are the post-ductal or preductal forms (proximal to the left subclavian artery) [23]. Coarctation of the aorta causes an increase in the blood pressure of the upper extremities. In its most severe presentation, it is associated with left ventricular failure after the closure of the ductus arteriosus, within the first one to two weeks after birth. Alternatively, it manifests in older children and adults, with upper extremity hypertension, leading to early coronary artery disease, aortic aneurysm, and cerebrovascular disease [24].

## 6. Aortic Age- and Gender-Related Changes

Aging is associated with a series of changes in aortic morphology and function, mostly stiffness and decreased distensibility (i.e., decreased dynamic change in the aortic cross-sectional area during the cardiac cycle) [25], and dilation. Stiffness is more pronounced in the abdominal aorta, while dilation involves all aortic segments, is more pronounced in the ascending aorta and aortic annulus (annuloaortic ectasia), and may cause aortic regurgitation. The morphologic substrate of age-related aortopathy includes fibrous intimal thickening, atherosclerosis, and medial degeneration (MD). The latter show features overlapping with those occurring in collagen diseases, albeit less extensive (see below, paragraph on the medial non-inflammatory degenerative disease). Age-related MD is a risk factor for aneurysms and AAD in older age. Recent evidence suggests that the impact of age on aortic dilation and stiffness may be worse in women than men [26,27,28]. Although women’s aortas are smaller than men’s, they tend to dilate faster with age [26]. A possible explanation is the greater extracellular matrix remodeling due to higher levels of matrix metalloproteinase in women as compared to males, leading to higher stiffness and reduced strength [29].

## 7. Thoracic Aortic Aneurysm

### 7.1. Definition and Epidemiology

A thoracic aortic aneurysm is defined as a permanent and localized dilation of the aorta (commonly defined as 1.5 times its normal size), involving all aortic layers (true aneurysm). The overall incidence of TAA is about 5 to 10 per 100,000 person-year. The majority of TAAs (60%) involve the aortic root and/or ascending aorta, followed by the descending aorta (40%) and the arch (10%). In 10% of cases, the whole thoracoabdominal aorta may be involved [30]. Rarely, dilation may selectively involve the aortic sinuses, usually the right. Most often, aneurysms of the sinuses are congenital lesions that may be associated with ventricular septal defects, less frequently acquired forms may be due to valve replacement complications, infective endocarditis, and syphilis [31,32].

Most patients with TAAs are asymptomatic and are diagnosed when an acute event occurs (aortic wall rupture, AAD, etc.). Although men show a higher frequency of TAAs, women show the worst outcome, as highlighted by higher rates of aortic growth (despite adjustment for body size), aortic rupture, and dissection at smaller aortic diameters and a higher rate of complications after surgery [26,28].

### 7.2. Etiology

The etiology of TAAs varies with location. Atherosclerosis is the most important risk factor for descending TAAs, as for aneurysms of the abdominal aorta. Instead, aneurysm of the aortic root and ascending aorta is associated with a combination of conditions that (i) increase aortic wall stress (such as systemic hypertension, cocaine abuse, coarctation of the aorta, etc.) and (ii) lead to structural abnormalities of the tunica media, such as degenerative/noninflammatory disease (including aging) or inflammatory disorders (atherosclerosis, infectious and non-infectious aortitis) [30]. Certain drugs, such as the fluoroquinolone class of antibiotics (ciprofloxacin, levofloxacin, and moxifloxacin) have been associated with both an increased incidence and adverse outcomes of aortic aneurysms and AAD [33,34], and thus, according to the Food and Drug Administration, should not be used by patients with high-risk conditions [35].

Generally, up to 25% of TAAs are familial. Familial TAAs can be syndromic (such as Marfan, Loeys–Dietz, Ehlers–Danlos, Turner syndrome, etc.) or non-syndromic (genetic, non-syndromic, familial TAAs). Altogether genetic forms have been associated with variants in genes that encode for proteins of extracellular matrix (ECM) (e.g., *FBN1, COL3A1*), TGF-β signaling pathway (e.g., *TGFB2*, *TGFBR1*, *TGFBR2*, and *SMAD3*), and contractile apparatus of smooth muscle cells (*ACTA2*, *MYH11, MYLK, PRKG1*) [36,37,38,39].

Aside from genetic, recent evidence suggests that a switch from oxidative to glycolytic metabolism could be a common driver of diverse TAAs [40]. Studies on patients with Marfan syndrome and experimental models of TAA demonstrate a decline of mitochondrial oxidative phosphorylation in aortic SMCs, induced by ECM disruption [41,42]. Although mechanisms linking ECM derangement to mitochondrial dysfunction in TAA are not completely understood, it has been shown that TGF-β signaling negatively regulates levels of PGC1alpha, the most important modulator of mitochondrial biogenesis [42]. Thus, vascular metabolism can be considered a new target to prevent AAS in patients with aortic dilation [43]. Notably, both mitochondria and ECM have been identified as potential targets of ciprofloxacin, suggesting metabolic imbalance as a possible mechanism involved in fluoroquinolone toxicity [44], as described with other classes of antibiotics [45].

As previously mentioned, aortic dilation has been reported in up to 40% of patients with BAV-related aortopathy. The latter can involve either the ascending aorta beyond the STJ (i.e., ascending phenotype, accounting for 70% of cases, typically presenting in adults with aortic valve sclerosis/stenosis) or the aortic root (i.e., sinuses of Valsalva and annulus, accounting for about 20% of cases, more frequent in younger males with moderate/severe aortic regurgitation). Extended phenotypes involving the root, the ascending aorta, and the arch are more rarely encountered [18]. The mechanisms underlying the development of thoracic aortic aneurysms in patients with BAV are not completely understood. Both significantly increased wall stress due to flow-related disturbances on the proximal aorta and genetic abnormalities resulting in histologic changes leading to aortic wall weakness are called into question. Rare or private mutations in several genes involving different molecular pathways, including neural crest migration, have been associated with BAV [15].

### 7.3. Morphologic Substrates of TAAs

The definition and classification of the morphological substrates of aortic diseases have been the subject of a thorough review by the Society for Cardiovascular Pathology (SCVP) and the Association for European Cardiovascular Pathology (AECVP) [46,47]. The updated classification system provides a standardized method for the pathology report of surgical specimens in inflammatory and non-inflammatory degenerative disease, enabling histopathologic features to correlate with clinical history and outcome in large case series. The most recent finding obtained by applying the SCVP/AECVP consensus on large case series will be briefly reviewed in the following paragraphs.

#### 7.3.1. Medial Non-Inflammatory Degenerative Disease

Medial degeneration is the most common morphologic substrate of TAAs, accounting for up to 67% of cases [48]. According to SCVP/AECVP classification system, individual components of MD are the following: (1) an increase of medial mucoid extracellular matrix (MEMA), which creates translamellar and/or intralamellar expansions; (2) elastic fibers fragmentation/loss; (3) elastic fibers thinning and disarray; (4) smooth muscle cell nuclear loss, which causes compaction of elastic fibers (laminar media collapse); and (5) collagen increase [47] (Figure 3A–C).

The degree of degeneration may vary from very little to severe, with focal multifocal or extensive distribution within the aortic wall [47].

According to large case series, patients with Marfan syndrome show a higher degree of medial degeneration and mucoid extracellular matrix accumulation as compared with other hereditary syndromes and BAV-aortopathy [49,50]. Generally, a higher degree of degenerative changes correlates with increased aortic diameter [50]. Immunohistochemical analysis on aortic samples from Marfan patients shows that aside from degenerative features, residual aortic SMCs are characterized by downregulation of the mitochondrial transcription factor TFAM and several mitochondrial respiratory chain subunits, as well as by increased expression of the glycolysis-related genes HIF1A and MYC, in line with the previously mentioned glycolytic rewiring [41].

#### 7.3.2. Inflammatory Aortic Disease

According to the AECVP and SCVP classification, there are three broad categories of inflammatory aortic disease, which in order of increasing inflammation are atherosclerosis, atherosclerosis with excessive inflammation, and aortitis/periaortitis [37].

Aortitis is increasingly recognized as a cause of TAAs with an incidence ranging from 4 to 25%, also considering atherosclerosis [48,51]. The latter account for up to 80% of AAA. Recently, Leone and colleagues, by analyzing 255 aortic samples from patients undergoing surgery for TAA, showed that, aside from medial degeneration, atherosclerosis (18.8%) and aortitis (13.7%) accounted for a significant portion of the samples [48]. Interestingly, up to 25% of patients had a mixed (degenerative-atherosclerosis, degenerative-aortitis, or atherosclerosis-aortitis) morphologic substrate, pointing to degeneration and inflammation of the aortic wall as not mutually exclusive phenomena. Compared with degenerative patients, atherosclerotic patients with ascending TAA were older and more frequently had a history of hypertension, hypercholesterolemia, diabetes, current smoking, a history of coronary artery disease, and frequently have a concomitant abdominal aortic aneurysm [48]. These results highlight the link between AAA and TAA, prompting further studies focused on the pathogenic role of inflammation, also in light of recent data showing that anti-inflammatory interventions in AAA could be potentially harmful and should be carefully monitored [52].

Aortitis and periaortitis refer to a wide spectrum of inflammatory disorders characterized by chronic inflammation restricted to the adventitia (i.e., periaortitis) or involving the media and/or intima (i.e., aortitis). Pathogenesis is often immune-mediated. Infectious causes are rare, and more often observed after invasive procedures or surgery (including aortic graft infections). Aortitis may be the presenting feature of systemic vasculitis or the unique manifestation of the disease, often identified only by histopathology. Aortitis is classified pathologically based on the pattern of the inflammatory infiltrate (i.e., granulomatous/giant cell, lymphoplasmacytic, suppurative pattern, etc.) (see Table 1).

Giant cell arteritis (GCA, Horton’s disease) is the most common vasculitis involving the aorta. It manifests in the elderly (peak incidence 70–80 years) with a male/female ratio of 1 to 3. The aortic involvement can precede, be concomitant, or follow the involvement of extra-cranial branches of the carotid arteries (especially the temporal superficial arteries). Histologically, GCA is characterized by non-compact granulomas in the media with macrophages and giant cells, associated with lymphoplasmacytic infiltrate (Figure 3D). Laminar medial necrosis is often observed, in line with the hypothesis that this morphologic finding likely represents the common result of different pathogenetic mechanisms, instead of a specific feature of degenerative disease. Intimal hyperplasia is frequently present, and, in an advanced stage, there may be extensive scarring [53].

Periaortitis is a common feature of IgG4-related disease, affecting about 20–35% of patients [54]. The abdominal aorta is more commonly affected, with dense lymphoplasmacytic infiltrates rich in IgG4-positive plasma cells and adventitial fibrosis. Less frequently (up to 8% of cases) IgG4-related disease manifests with TA aortitis and TAA [55,56]. The IgG4-related disease accounts for about 75% of all lymphoplasmacytic aortitis.

Among infectious aortitis, it is worth mentioning vascular Q fever, a zoonotic disease caused by Coxiella burnetii. Most patients show mild or asymptomatic acute infections. However, an estimated 1–5% progress to a chronic phase. Vascular infections are the second most common form of chronic Q fever, following endocarditis. Although it has been observed more often in the AA, infections have also been reported in the TA and vascular graft [57].

## 8. Acute Aortic Syndrome

Acute aortic syndromes are a group of life-threatening conditions that include AAD, intramural hematoma, aortic pseudoaneurysm, and traumatic aortic injury affecting the aortic wall. They are characterized by similar clinical presentations and share diagnostic and interventional approaches (recently reviewed in 30).

### 8.1. Thoracic Aorta Dissection

#### 8.1.1. Definition and Epidemiology

Aortic dissection, particularly when involving the ascending aorta (type A in Stanford classification), is a life-threatening condition comprising 80–90% of all AAS. An intimal tear is usually the initiating condition, resulting in the destruction of the media by blood flow and creating a false lumen. This process is followed either by an aortic rupture in case of adventitial disruption or by reconnection of the false lumen with the true lumen through a second intimal tear.

Due to the lack of comprehensive prospective population studies and the decreased number of autopsies performed worldwide, epidemiological data on AAD are still deficient. The suggested incidence ranges from 2.9 to 9.1 per 100,000 person-year [30,58,59]. Most patients with AAD are male in the seventh decade of life; affected women are older than men and show higher in-hospital mortality [27,28].

#### 8.1.2. Risk Factors and Morphologic Substrates

Risk factors and morphologic substrate of AAD overlay with those of TAA. Systemic hypertension, atherosclerosis, and iatrogenic causes are more prevalent in older patients, whereas inherited conditions (first Marfan syndrome) are typical of younger patients. The prevalence of BAV in patients with AAD appears higher as compared to the general population [60].

Medial degeneration is the most common substrate of AAD, with elastic fibers fragmentation and thinning being the most observed features, followed by MEMA. According to a few studies which attempt to discriminate histopathologic features between TAA and AAD, the degree of elastic fibers fragmentation and translamellar MEMA is significantly increased in TAD versus TAA [41]. In addition, laminar media collapse is much more frequent and extensive in the acute setting [61]. Notably, although a higher score of MD is usually observed in dilated aortas, degenerative changes have been described even in small aortae with AAD (diameters <55 mm), in line with the concept that risk stratification of aortic dissection based on aortic dimensions is imperfect [27,28]. Similar to what has been described for TAA, moderate/severe atherosclerosis may coexist with medial degeneration in up to 25% of cases, especially in older patients with cardiovascular risk factors [61].

### 8.2. Acute Aortic Syndromes Other Than AAD

Aortic intramural hematoma is characterized by hemorrhage within the aortic wall in the absence of an intimal flap, a false lumen, or a primary intimal tear. It is a dynamic condition which may progress to classic AAD or aortic rupture, or even spontaneously regress [30]. The hematoma is typically located in the media but can extend right below the adventitia. The pathophysiology of this condition is still to be defined with certainty, but the leading hypotheses point to the vasa vasorum as the possible culprit. These vessels could rupture following acute traumatic stress or spontaneously bleed in a setting of pathological neovascularization, analogously to what is seen in atherosclerotic plaque rupture [30,62]. Penetrating aortic ulcer is an ulceration of an aortic atherosclerotic plaque penetrating the internal elastic lamina into the media, usually in a setting of widespread atherosclerosis of the thoracic aorta. Penetrating aortic ulcers can be single or multiple and can eventually stabilize, present minimal growth over years, or evolve to intramural hematoma or transmural rupture [30]. Histologically, penetrating aortic ulcers present with intimal degeneration, cholesterol deposition, and medial involvement, often taking the form of cystic medial necrosis [63]. Both intramural hematoma and penetrating ulcers are more common in the descending aorta as compared to ascending aorta and arch. Risk factors are analogous to those observed in AAD, although patients tend to be older. Pseudoaneurysms are dilations of the aorta due to disruption of all the aortic wall layers, enclosed only by the periaortic connective tissue [30].

## 9. Conclusions

In conclusion, the present article offers an overview of aortic disease based on morphologic substrates and embryologic development. The complex embryology of the aortic root and aortic arch accounts for the congenital anomalies that develop at these sites, which can be asymptomatic or manifest clinical consequences even in older ages, as is seen with BAV. The heterogeneous histogenesis of the tunica media contributes to explaining the development of degenerative and inflammatory disorders prevalently within the TA. Histologic analysis of large case series, by applying the AECVP and SCVP classification system, highlights new clinicopathological correlations pointing to the importance of histological analysis of aortic samples to better stratify patients in the long-term follow-up, after surgical treatment. In addition, the histologic analysis highlights the relationship between inflammation and degeneration, suggesting complex pathophysiological mechanisms that need to be fully addressed in the future. Finally, recent findings point to metabolic reprogramming as a common driver in aortic aneurysm development and progression, paving the way to new therapeutic opportunities.

## Figures and Tables

**Figure 1 diagnostics-13-02166-f001:**
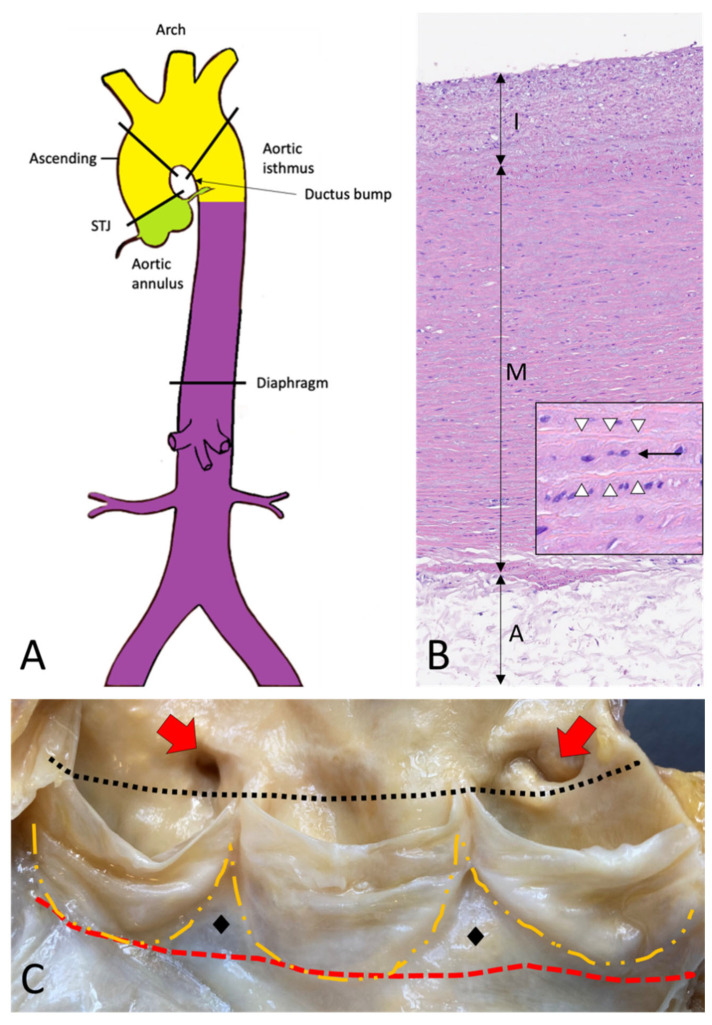
Topographic anatomy and normal histology of the aorta. (**A**)—Diagram of the aorta with different colors reflecting the different histogenesis of the SMCs of the tunica media: SMCs of the aortic root derive from the secondary heart field (lateral plate mesoderm) (green); SMCs of ascending aorta and the arch derive from the neural crests (ectoderm) (yellow); SMCs of the descending and abdominal aorta derive from the somites of the paraxial mesoderm (purple). SMCs = smooth muscle cells. (**B**)—Histological anatomy of the descending aorta with tunica intima (I), media (M), and adventitia (A). The mild fibrous intimal thickening is a typical age-related change. The normal lamellar architecture of the tunica media is preserved and showed at higher magnification in the insert, with its elastic laminae (white arrowheads) and the intervening smooth muscle cells (black arrow) (Hematoxylin and eosin, original magnification 10×; insert: 40×). (**C**)—The aortic root is the segment between the annulus (red dashed line) and the sinotubular junction (black dotted line) and comprises the sinuses of Valsalva with the aortic cusps (outlined with the orange dashed line), the origin of the coronary arteries (red arrows), and the intercuspal triangles (black diamonds).

**Figure 2 diagnostics-13-02166-f002:**
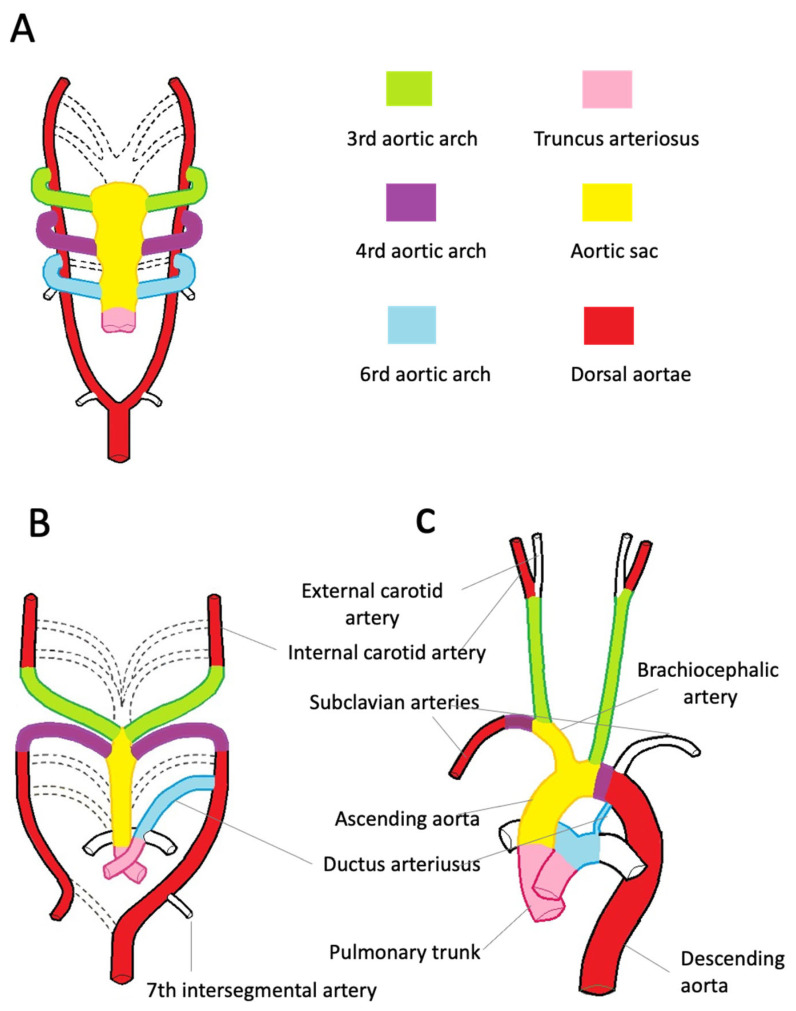
Embryologic development of the thoracic aorta and its branches. (**A**)—The scheme highlights the development of the six arches from the aortic sac during the 5th week of development, the 1st and 2nd arches have regressed (dashed line). The truncus arteriosus is partially divided by the conotruncal septum. (**B**)—Aortic arches at 7 weeks of development. The segments of the dorsal aorta connecting the third and fourth arch arteries disappear, as well as the fifth, part of the right and sixth arches and a portion of the right dorsal aorta. (**C**)—At 8 weeks, the thoracic aorta and the epiaortic vessels have almost completed their development. Note the patency of the ductus arteriosus, originating from the left sixth aortic arch.

**Figure 3 diagnostics-13-02166-f003:**
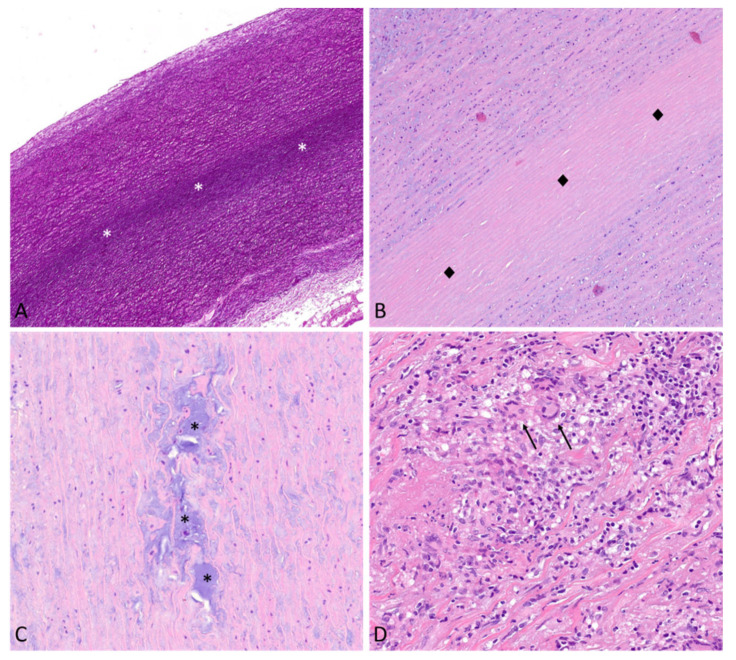
The morphologic substrates of thoracic aortic aneurysms. (**A**)—Extensive, thin, laminar medial collapse (white asterisks). Elastic fibers collapse forming a thin band (Elastic Van Gieson stain, original magnification 10×). (**B**)—Hematoxylin and eosin stain highlights SMCs nuclear loss within laminar collapse (black diamonds). Mild intralamellar mucoid extracellular matrix accumulation can be observed (Hematoxylin and eosin, original magnification 20×). (**C**)—Abundant intralamellar and translamellar (black asterisks) mucoid extracellular matrix accumulation (Hematoxylin and eosin, original magnification 20×). (**D**)—Giant cell aortitis. The aortic wall structure is deranged because of numerous epithelioid macrophages and giant cells (arrows) arranged in non-compact granulomas. Lymphocytes, plasma cells, and rare granulocytes are also present (Hematoxylin and eosin, original magnification 20×).

**Table 1 diagnostics-13-02166-t001:** Histologic patterns of arteritis.

Granulomatous/Giant Cell	
Non-infectious Giant cell arteritis Takayasu’s arteritis Granulomatosis with polyangiitis Sarcoidosis Rheumatoid arthritis	Infectious Syphilitic aortitis Mycobacterial Fungal infection
**Lymphoplasmacytic**	
Non-infectious IgG4-related disease Ankylosing spondylitis Systemic lupus erythematosus	Infectious Syphilitic aortitis
**Suppurative**	
Non-infectious	Infectious Staphylococcus, Streptococcus, Salmonella, Pseudomonas, fungal infections
**Mixed**	
Non-infectious Behçet’s disease	Non-infectious

## Data Availability

No new data were created or analyzed in this study. Data sharing is not applicable to this article.

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
