# Peer review of "Thoracic Aorta: Anatomy and Pathology"

_diagnostics, 2023, doi:10.3390/diagnostics13132166_

Round 1
Reviewer 1 Report
This is an excellent brief, but highly focused, review of the thoracic aorta – anatomy, embryology, histopathology and disease classification. A particular strength of the review is its detailed inclusion of the recent literature, particularly in relation to recent consensus statements on the histopathological description of thoracic aortic disease.
Overall, I have no significant issues with the review. The figures and tables are well presented illustrations of the content s of the text.
A few minor issues that the authors may wish to address prior to publication are listed below:
1. Line 181: “ ….during the first month of LIFE.”
2. Line 204: remove mid-sentence return
3. Line 219: while definitions of TAA size vary, a common definition is 1.5 times its normal size.
Author Response
This is an excellent brief, but highly focused, review of the thoracic aorta – anatomy, embryology, histopathology, and disease classification. A particular strength of the review is its detailed inclusion of the recent literature, particularly in relation to recent consensus statements on the histopathological description of thoracic aortic disease. Overall, I have no significant issues with the review. The figures and tables are well presented illustrations of the contents of the text.
We thank the Reviewer for his/her encouraging and supportive comment.
A few minor issues that the authors may wish to address prior to publication are listed below:
- Line 181: “ ….during the first month of LIFE.”
We modified the text accordingly.
- Line 204: remove mid-sentence return.
We thank the Reviewer for pointing out our imprecision. We removed mid-sentence return.
- Line 219: while definitions of TAA size vary, a common definition is 1.5 times its normal size.
We thank the Reviewer for his/her suggestion. We added a short sentence.
Reviewer 2 Report
The authors tried to present a series of aortic disease based on morphologic substrates and embryologic development. There are several issues:
1 The informations presented are very well known facts and the paper doesn't present any new insights
2 They are many pathologies presented and because of this a subject like bicuspid aortic valve is presented in a superficial fashion, witch is the case off all pathologies presented
3 Linking morphology and embryology to the aortic disease is not something new and can be seen in a lot of publications
English Language is used accordingly
Author Response
The authors tried to present a series of aortic disease based on morphologic substrates and embryologic development. There are several issues:
1 The information presented are very well-known facts and the paper doesn't present any new insights.
We apologize for this limitation. We have included several new themes and references in the revised version of the manuscript. We hope this will make it more appealing to the Reviewer.
2 They are many pathologies presented and because of this a subject like bicuspid aortic valve is presented in a superficial fashion, which is the case of all pathologies presented
We apologize for this deficiency. We were asked to write a review on embryological and pathological aspects of the thoracic aorta to be included in a special issue devoted to this organ. The special issue already includes articles on specific pathologies such as the bicuspid aortic valve. For this reason, we organized our manuscript as an overview of diseases of the aorta, limiting our presentation to short paragraphs reporting on more recent literature data. We have added new topics that we hope will enrich the overall quality of the work.
3 Linking morphology and embryology to the aortic disease is not something new and can be seen in a lot of publications.
We agree with the Reviewer. However, we feel that in a manuscript devoted to embryology and anatomy of the aorta, this concept should be reiterated.
Reviewer 3 Report
Thoracic Aorta: anatomy and pathology
While in general it is a nice overview of the anatomy and pathology of the thoracic aorta, I do have some questions/remarks. And I also wonder what the novelty is.
“In the TA, vasa vasorum normally extend into the outer third of the media. Vasa vasorum are absent in the AA.” From the AA and AAA sections I have studied I can say that the last sentence is not true. Please remove this statement.
“Altogether genetic forms have been associated with variants in genes that encode for proteins of extracellular matrix (e.g., FBN1)” It would be good to also mention COL3A1 here, since collagens are just as important fibrillar proteins as fibrillin-1.
“Histopathologic features of MD have been observed in cases of inflammatory disease of the aorta, with higher scores than age-matched controls, suggesting that degeneration and inflammation of the aortic wall are not mutually exclusive, and that inflammatory damage might play a role in inducing medial degenerative changes [42]. ” Somehow it seems as if the authors are surprised about this? That is how it is framed now. Yet, the purpose of inflammation is obviously repair. Yet, an optimal inflammatory response is necessary for repair and if this is out of balance, inflammation may indeed contribute to disease. This is controlled by the local make-up of the different phenotypic inflammatory cells. It is a sterile inflammation here. In contrast to later mentioned infections that can contribute to aorta pathology. Perhaps this can be explained a bit better.
“7.3.2. Inflammatory aortic disease ” Is it of interest to also mention chronic vascular Q fever in the aorta? It has been observed more often in the AA, but also in the TA and vascular graft. (doi: 10.7554/eLife.72486. AND doi: 10.1177/1538574416642876. AND doi: 10.1093/ejcts/ezs217)
In relation to inflammation (and to create some discussion) it may also be of interest to mention certain drugs that are perhaps detrimental in patients with underlying aortic disease, such as the antibiotic class of fluoroquinolones that are contraindicated in aneurysm patients by causing mitochondrial dysfunction. Similarly, the tetracyclines influence the mitochondria, which may be helpful using the drugs short term to decrease the inflammatory cell activity, yet perhaps contributing to vascular disease using them long term (doi: 10.3390/ijms22084100. AND doi: 10.1016/j.celrep.2015.02.034.). Along the same line, immunosuppressive medication may not only be harmless in aortic disease (doi: 10.1161/CIRCULATIONAHA.110.008573.), for similar reasons that they do not just influence inflammatory cells, but often also all other mammalian cells when used chronically. Since in Marfan syndrome and AAA it has been established that intrinsic mitochondrial dysfunction is present in the aortic tissue (mostly in the smooth muscle cells and the cause may be different in different aneurysmal diseases) (refs: J Am Heart Assoc. 2021 Sep 7;10(17):e020231. doi: 10.1161/JAHA.120.020231. AND Circulation. 2021 May 25;143(21):2091-2109. doi: 10.1161/CIRCULATIONAHA.120.051171. AND Arterioscler Thromb Vasc Biol. 2022 Apr;42(4):462-469. doi: 10.1161/ATVBAHA.121.317346. AND Br J Pharmacol. 2023 Mar 25. doi: 10.1111/bph.16077. ), this is likely why the aneurysmal cells are sensitive to these drugs. Thus apart from inflammation, the metabolic reprogramming or cellular exhaustion is considered now as key target for intervention.
Minor: The numbers in Figure two are rather small. Perhaps they can be put next to the cartoon instead of in the cartoon of the aortic development images? Or make the images larger and then the two first above and the third underneath with the legend boxes next to it?
“cross-sectional area during (gap) the cardiac cycle) [25], and dilation.” There is a gap in the middle of this sentence.
Author Response
- While in general, it is a nice overview of the anatomy and pathology of the thoracic aorta, I do have some questions/remarks. And I also wonder what the novelty is.
We strongly thank the reviewer for his/her encouraging and supportive comments. As suggested, we have included several new themes and references in the revised version of the manuscript. We hope this will make it more interesting.
- “In the TA, vasa vasorum normally extend into the outer third of the media. Vasa vasorum are absent in the AA.” From the AA and AAA sections I have studied, I can say that the last sentence is not true. Please remove this statement.
We apologize to Reviewer for the incorrectness, maybe we should have said that vasa vasorum in the AA are decreased in number as compared to the TA. We removed the sentence as suggested.
- “Altogether genetic forms have been associated with variants in genes that encode for proteins of extracellular matrix (e.g., FBN1)” It would be good to also mention COL3A1 here, since collagens are just as important fibrillar proteins as fibrillin-1.
We thank the Reviewer for his/her suggestion. We added a mention of COL3A1 in the text.
- “Histopathologic features of MD have been observed in cases of inflammatory disease of the aorta, with higher scores than age-matched controls, suggesting that degeneration and inflammation of the aortic wall are not mutually exclusive and that inflammatory damage might play a role in inducing medial degenerative changes [42]. ” Somehow it seems as if the authors are surprised about this? That is how it is framed now. Yet, the purpose of inflammation is obviously repair. Yet, an optimal inflammatory response is necessary for repair and if this is out of balance, inflammation may indeed contribute to disease. This is controlled by the local make-up of the different phenotypic inflammatory cells. It is a sterile inflammation here. In contrast to later mentioned infections that can contribute to aorta pathology. Perhaps this can be explained a bit better.
We thank the reviewer for his comment. We agree that unbalanced inflammation can be dangerous, either in the sense of hyper immunity (see aortitis) or immunosuppression (see immunosuppressive regimen and aneurysm progression). However, we wanted to report the findings of Leone and colleagues (J Thorac Cardiovasc Surg. 2020;160(6):1434-1443.e6. doi:10.1016/j.jtcvs.2019.08.108) showing that, in addition to medial degeneration, atherosclerosis, and aortitis are common substrates of TAA and are often observed in combination with MD. In these cases, degeneration of the media could be the result of aging and could be aggravated by atherosclerosis (and associated inflammation). This concomitance of events, in subjects with a specific metabolic phenotype (hypertension, hypercholesterolemia, diabetes, current smoking, history of coronary artery disease) might lead to the development of TAA. Considering these findings, it is even more important to urge further studies on the pathogenetic role of inflammation, also in view of the data you cited, which demonstrate that anti-inflammatory interventions can be potentially harmful and should be carefully monitored. We have modified the text in the new version of the manuscript; we hope it is now clearer.
- “7.3.2. Inflammatory aortic disease ” Is it of interest to also mention chronic vascular Q fever in the aorta? It has been observed more often in the AA, but also in the TA and vascular graft. (doi: 10.7554/eLife.72486. AND doi: 10.1177/1538574416642876. AND doi: 10.1093/ejcts/ezs217)
We thank the Reviewer for this important suggestion. We added a few sentences and a reference in the revised version of the manuscript.
- In relation to inflammation (and to create some discussion), it may also be of interest to mention certain drugs that are perhaps detrimental in patients with underlying aortic disease, such as the antibiotic class of fluoroquinolones that are contraindicated in aneurysm patients by causing mitochondrial dysfunction. Similarly, the tetracyclines influence the mitochondria, which may be helpful using the drugs short term to decrease the inflammatory cell activity, yet perhaps contributing to vascular disease using them long term (doi: 10.3390/ijms22084100. AND doi: 10.1016/j.celrep.2015.02.034.). Along the same line, immunosuppressive medication may not only be harmless in aortic disease (doi: 10.1161/CIRCULATIONAHA.110.008573.), for similar reasons that they do not just influence inflammatory cells, but often also all other mammalian cells when used chronically. Since in Marfan syndrome and AAA it has been established that intrinsic mitochondrial dysfunction is present in the aortic tissue (mostly in the smooth muscle cells and the cause may be different in different aneurysmal diseases) (refs: J Am Heart Assoc. 2021 Sep 7;10(17):e020231. doi: 10.1161/JAHA.120.020231. AND Circulation. 2021 May 25;143(21):2091-2109. doi: 10.1161/CIRCULATIONAHA.120.051171. AND Arterioscler Thromb Vasc Biol. 2022 Apr;42(4):462-469. doi: 10.1161/ATVBAHA.121.317346. AND Br J Pharmacol. 2023 Mar 25. doi: 10.1111/bph.16077.), this is likely why the aneurysmal cells are sensitive to these drugs. Thus, apart from inflammation, the metabolic reprogramming or cellular exhaustion is considered now as key target for intervention.
We warmly thank the Reviewer for his/her significant suggestion. We agree we the concept that mitochondrial dysfunction and metabolic switch are emerging as important drivers of aneurysm development in the thoracic aorta, as already demonstrated in the abdominal aorta. We also agree that the detrimental function of fluoroquinolones may be attributed to mitochondrial damage. We added a few sentences and references on this topic in the new version of the manuscript. We also reported, in the paragraph on morphologic substrates, results relative to immunostaining on human aortic samples, strengthening the role of metabolic rewiring in aneurismal progression (J Am Heart Assoc. 2021 Sep 7;10(17):e020231. doi: 10.1161/JAHA.120.020231).
- The numbers in Figure two are rather small. Perhaps they can be put next to the cartoon instead of in the cartoon of the aortic development images? Or make the images larger and then the two first above and the third underneath with the legend boxes next to it?
We thank the Reviewer for his/her suggestion. A new figure on the embryology of the aorta has been provided in the revised version of the manuscript.
- “cross-sectional area during (gap) the cardiac cycle) [25], and dilation.” There is a gap in the middle of this sentence.
We removed the gap.
Reviewer 4 Report
It is advisable to better illustrate the section of congenital pathology of the aorta.
I would recommend to complete the section on intramural hematoma and penetrating aortic ulcer
Author Response
We thank the Reviewer for his/her suggestions.
- It is advisable to better illustrate the section of congenital pathology of the aorta.
A revised figure on embryology of the aorta has been provided in the revised version of the manuscript.
- I would recommend completing the section on intramural hematoma and penetrating aortic ulcer.
A few sentences on intramural hematoma and penetrating aortic ulcer have been added in the in the revised version of the manuscript.
Round 2
Reviewer 2 Report
Although very well organized almost all the information can be found in different sources. Perhaps focusing only one pathology (eg bicuspid aortic valve) can improve the quality of the paper.
English is used accordingly
Author Response
- Although very well organized almost all the information can be found in different sources.
We thank the Reviewer for his/her very positive comment.
We agree that all information can be found in different sources. However, for the first time, we conduct a literature review focusing on the use of the new AECVP and SCVP consensus statement on histopathological diagnostic criteria for aortic disease.
In the past 3-4 years, the application of the new classification system by several Authors has highlighted new concepts that we summarize here. For example, although MD is confirmed as the most common finding in ascending TAA, atherosclerosis, and aortitis are quite common; mixed forms are quite common, especially in the elderly. Medial lamina collapse is the most severe medial alteration, is more frequent and extensive in the acute setting, and is probably the common result of several pathogenetic mechanisms (degenerative and inflammatory). Aortitis is responsible for degenerative changes that appear more severe than aging or other degenerative processes.
Overall, these results underscore the usefulness of the new classification system, suggesting new clinicopathologic correlations. They also underscore the importance of histologic analysis of aortic specimens to better stratify patients in long-term follow-up after surgical treatment.
We believe this represents an important message for clinicians and scientists.
New findings regarding the role of mitochondrial dysfunction also show parallels with the morphologic changes mentioned in the manuscript, underscoring the importance of histology and morphologic analysis of surgical specimens, including using immunohistochemical methods.
We added a few sentences in the conclusion sections to better clarify our thought.
- Perhaps focusing only one pathology (eg bicuspid aortic valve) can improve the quality of the paper.
We thank the Reviewer for his/her comment.
We agree that dealing with many topics does not allow one to delve into any of them. For this reason, we have merged the paragraphs devoted to malformations of the arch of the aorta and aortic coarctation and given more space to the discussion of the bicuspid aorta. We favored information related to morphologic variants and anatomic-clinical correlations, in line with the topic of our manuscript.
We hope these changes improved the quality of our manuscript.
Round 3
Reviewer 2 Report
The authors have improved the manuscript and I recommed it for publication.
The english language used is correct.